# The Relationship between Subjective Aging and Cognition in Elderly People: A Systematic Review

**DOI:** 10.3390/healthcare11243115

**Published:** 2023-12-07

**Authors:** Óscar Fernández-Ballbé, Marina Martin-Moratinos, Jesus Saiz, Lorena Gallardo-Peralta, Ana Barrón López de Roda

**Affiliations:** 1Faculty of Psychology, Complutense University of Madrid, 28223 Madrid, Spain; jesus.saiz@psi.ucm.es (J.S.); logallar@ucm.es (L.G.-P.); abarronl@ucm.es (A.B.L.d.R.); 2Universitary Hospital Puerta de Hierro, 28222 Madrid, Spain; mmmoratinos27@gmail.com

**Keywords:** subjective aging, self-perceptions of aging, attitudes towards own aging, cognition, elderly adults, systematic review

## Abstract

There is a growing body of evidence on the effects of subjective aging on health, well-being and quality of life. This review aims to synthesize findings about the link between subjective aging and cognition and cognitive decline. Furthermore, it provides an examination of variation sources such as subjective aging construct, cognitive domains, measures employed, age and moderator variables. A systematic search was performed in PubMed, PsychInfo and Web of Science, as well as grey literature searches in Google Scholar, OpenGrey, WorldCat and NDLTD, which resulted in 59 reports being included. Subjective aging is a relevant construct in the explanation and prediction of cognitive aging and cognitive decline in elderly adults. More positive views about own aging and self-perceptions of aging, as well as a younger subjective age, were consistently related to better cognition and lower risk of cognitive decline. However, there were differences due to subjective aging subdimensions and cognitive domains, as well as an effect of age. Additionally, there were concerns about the content validity of some measures employed, such as the Philadelphia Geriatric Center Morale Scale for subjective aging and the Mini Mental State Examination for global cognition. Further studies should employ longitudinal designs with a process-based approach to cognition and precise subjective aging measures.

## 1. Introduction

The progressive aging of the population is one of the challenges of contemporary societies, implying attainment of adequate levels of health and quality of life in the elderly population [1,2]. While chronological age remains as a relevant predictor of aging trajectories, the progressive inclusion of social and psychosocial factors based on the active aging paradigm offers a multidimensional and more complete view [3,4].

Subjective aging is a relevant psychosocial variable in the prediction of health outcomes at elderly age. It is defined as the representations that arise from the subjective interpretation given to the personal process of aging and the attributions about one’s own age. These representations encompass several elements related to the process of aging. Following the reconceptualization proposed by Barker et al. [5] of the common-sense model of regulation [6], elderly people can perceive different degrees of chronicity, control, consequences and the emotional impact of them. For instance, elder people might perceive that aging has positive and negative consequences and that they do or do not have personal control over them. Moreover, this impact can be viewed as something chronic or may arouse different emotional reactions.

Once these representations are formed, they might lead to different outcomes in health and quality of life. The stereotype embodiment theory [7] proposes three possible pathways. The psychological pathway explains how self-fulfilling prophecies about the nature, consequences, and control of the aging processes impact behavior. For instance, there is evidence of the link between negative representations and lower self-efficacy related to perceiving physical losses as out of control and having, as a consequence, a negative impact on physical health [8]. Moreover, elderly people who engage in negative representations tend to use selection, optimization and compensation strategies to a lesser degree, specifically if they endorse a belief that physical losses are an inherent consequence of aging [9]. The behavioral pathway explains how negative representations are related to several healthy behaviors that impact health, such as smoking [10], lack of adherence to pharmacological treatments [11], and physical activity levels [12]. Lastly, the physiological pathway is related to autonomous nervous activation related to subjective aging. For example, elderly people with higher levels of negative representations show enhanced reactivity to stress, which increases the probability of a cardiac event [13].There is substantial evidence of the relationship between subjective aging and health outcomes in elderly people. Positive representations are linked to more years of life [14], higher functional health [15] and higher physical function [16], whereas negative representations are linked to a steeper decline in physical [17], and mental health [18]. Meta-analytic approaches indicate a small but consistent effect of subjective aging on health and mortality [19] as well as on subjective well-being and depression [20].

This review focuses on exploring the relationship between subjective aging, cognitive functioning, and the probability of developing mild cognitive impairment or dementia. As the cognitive enrichment theory [21] proposes, elderly people are able to modify their cognitive trajectories through healthy behaviors and lifestyles. Even as cognitive efficacy is strongly associated with age because of biological and neurological factors, there is a substantial degree by which psychosocial factors, such as subjective representations of aging, can modify them. In theoretical terms, it can be argued that the content of this representation includes cognitive characteristics, such as ‘distracted’, ‘forgetful’ and ‘wise’ [22]. These representations impact healthy behaviors, active lifestyles and selection, and optimization and compensation strategies, which, in turn, might lead to lower cognitive performance and a higher risk of developing neurodegenerative diseases [23,24]. Moreover, negative representations are linked to variability in brain structures that support the main cognitive functions, such as lower hippocampal volume [25] and inferior grey matter volume in the inferior frontal gyrus and the superior temporal gyrus [26]. There have been efforts to synthesize evidence regarding subjective age and cognition, finding a small but significant effect [20]; however, this systematic review aims to explore the complete umbrella concept of subjective aging as well as to address the potential sources of variation in this field.

First, there might be a disparity regarding the subjective aging construct addressed. There have been several conceptualizations employed, which are intertwined but do not reflect the same contents. Subjective age refers to the age someone feels he has, and it has usually been compared with chronological age [27]. However, age identity is defined as the degree of correspondence between felt age and age associated with the peer group or social role [28]. Self-perceptions of aging are representations that arise from the personal experience of aging [29], whereas attitudes towards own aging express self-directed cognitions, emotions and behaviors based on belonging to a certain age group [7]. Moreover, the concept of awareness of aging is defined as the experiences that relate to changes associated with aging [30]. Each of these constructs might impact differentially in the cognitive trajectories of elderly people. Theoretically, there might be differences between the pathways through which these constructs exert their effect. For instance, Westerhof & Wurm [31] proposed that age identity may contribute mainly to the psychological pathway since this construct is more closely related to dispositional variables such as optimism and self-esteem, whereas attitudes towards own aging might influence cognitive and behavioral pathways. This reasoning follows the current approaches to subjective aging, which considers the differences between highly aggregated measures of aging representations such as subjective aging and more specific and multidimensional approaches, as well as the specific processes involved in their relationships with health [32]. Moreover, a matching effect [33] between the valence or content of the representation of aging and the health outcome has been proposed. For example, Spuling et al. [34] found that self-rated health is more strongly predicted by the specific physical losses dimension of self-perceptions of aging when compared with broader constructs such as subjective age and attitudes towards own aging. A similar effect was found by Sabatini et al. [35], showing that the awareness of age-related changes losses dimension predicts cognition but not general attitudes towards own aging. This effect may be explained by the inclusion of specific cognitive items in the losses dimension, whereas attitudes towards own aging reflect general representations. Finally, there is evidence that some subjective aging constructs are more consistent when predicting several cognitive domains, such as awareness of age-related changes when compared with attitudes towards own aging [36].

A second source of variation is the measures employed to assess subjective aging. Even if there is a wide spectrum of questionnaires, most of them have not been designed to measure these constructs [37], and some scales include items referring both to general beliefs about aging and to personal representations [38]. Moreover, these measures differ in dimensionality. For instance, the unidimensional Philadelphia Geriatric Center Morale Scale (PGCMS; [39]) is one of the most common measures employed to assess attitudes towards own aging [37], albeit there is consensus about the construct multidimensionality [40].

Other relevant variables are age and the cognitive construct evaluated. Although there is evidence showing a decrease in cognitive efficiency that commences in the later years of middle age, there is some variability depending on the cognitive domain. For instance, speed of processing and inhibition start to show decreased efficiency during the early years of old age [41], and they secondarily affect some complex cognitive processes such as alternating attention and working memory update [42]. However, domains such as language and simple attentional processes do not show this effect until reaching old age [43]. The specific measures employed to measure such constructs may also affect the results since they might reflect subcomponents of the cognitive domain or not be the ideal candidate to measure it. In addition, there might be an interaction with an age effect on subjective aging since this variable accounts for a larger effect on midlife and young old samples than in elderly samples [19].

Finally, this review aims to examine potential moderator and mediator variables. Regarding physical mental health, several constructs have been proposed, such as self-efficacy [8], healthy behaviors [44], leisure activities [45] and loneliness [46]. Since the stereotype embodiment proposes an effect through psychological, behavioral and physiological pathways, it is particularly relevant to address the variables through which subjective aging exerts an effect on cognition. One possible moderator variable regarding subjective aging and cognition is loneliness. There is substantial evidence about the detrimental effects of loneliness on cognition [47] and cognitive decline [48]. Moreover, subjective aging seems to play a relevant role in the relationship between loneliness and other health outcomes. For instance, subjective age moderates the effect of loneliness and psychological symptoms [49], and loneliness moderates the relationship between self-perceptions of aging and depressive symptoms [46]. Therefore, it is plausible that loneliness interacts with subjective aging through a psychological pathway, which, in turn, impacts cognition. Another candidate is depression. This condition has a notable effect on cognition and dementia risk [50], and it is a critical key point when diagnosing dementia [51]. Moreover, subjective age is a significant predictor of depression [20], and that relationship seems to be unidirectional [52]. Thus, it is possible that subjective age impacts depression, which, in turn, may affect cognition via behavioral and psychological pathways. Finally, there are reasons to consider physiological pathways. For instance, inflammation is a relevant variable for dementia [53], and it has shown a moderator effect between self-perceptions of aging and longevity [54].

## 2. Method

### 2.1. Search Strategy

The main searches were conducted in PubMed, Web of Science and PsychInfo. Additional searches for grey literature were performed in Google Scholar, OpenGrey, WorldCat and NDLTD in order to reduce the risk of publication bias. Searches were last performed on 9th September 2023. The complete syntax employed in each database is available in Appendix A.

To determine all the constructs under the umbrella term of subjective aging, the first researcher detected the main terms for subjective aging based on Diehl & Wahl [30]: subjective age, age identity, self-perceptions of aging, attitudes towards own aging and aging self-awareness. Then, he searched for synonyms and related constructs, resulting in a total of 21 relevant terms. Most of the terms specified are included in recent systematic reviews and meta-analyses about subjective aging [19,20]. The complete list of terms is available in Appendix A. This review employs the PRISMA system) and was registered in PROSPERO (CRD42023429916).

### 2.2. Inclusion and Exclusion Criteria

The inclusion criteria for studies in this systematic review are (a) ex post facto designs, (b) published in English or Spanish, (c) to include at least one measure of subjective aging, (d) to include at least one objective measure of cognition or cognitive decline and (e) to include a sample of 50 years old or elderly at any measurement point. These criteria try to ensure that all evidence regarding cognition and subjective aging is included, independently of whether it comes from a cross-sectional or a longitudinal design. Even if longitudinal designs are more suited to obtaining high-quality and semi-causal conclusions, evidence from cross-sectional designs is also valuable information. Moreover, the inclusion of samples of 50 years or elderly warrants that some normotypic changes in cognition associated with age are in play. Studies that (a) employed an ad hoc measure of subjective aging that is not described in detail, (b) measured objective cognition with an ad hoc measure that is not described in detail or (c) did not employ statistical analysis were excluded.

### 2.3. Study Selection

The study selection was performed using the Rayyan platform. First, OFB and MMM included all the studies found in the database searches and eliminated duplicates. Then, OFB and MMM independently selected potential candidates for inclusion by analyzing title and abstract. Disagreements were resolved through consensus. In a second phase, OFB and MMM independently selected the final articles included by performing a full-text review. Disagreements were resolved through consensus and assessment by a third author (JSG).

### 2.4. Data Extraction

Data extraction included (a) citation, (b) publication type, (c) purpose, (d) design, (e) sampling, (f) sample characteristics, (g) subjective aging construct and measures, (h) cognitive domain and measures, (i) moderators and measures and (j) results. The specific statistical results obtained (e.g., partial correlations, r2 adjusted, standardized regression coefficients, Cohen’s d) are described in the Results section.

### 2.5. Quality Assessment

Quality assessment was performed using a modified version of the Newcastle–Ottawa Scale, available in Appendix A. It includes the following elements: (a) if the sample is representative of 50 years old or elderly population, (b) if the sample is extracted from such a representative population, (c) the presence of at least a measure of subjective aging, (d) the presence of at least a measure of cognition or neurodegenerative disease, (e) if its statistical analysis permits the examination of the relationship between subjective aging and cognition and (f) if it controls relevant variables through inclusion and exclusion criteria or by including them in the statistical analysis. Ratings vary between 1 and 6, 1–2 being an indicator of low quality, 3–4 medium quality, and 5–6 high quality.

## 3. Results

### 3.1. Search Results

The results obtained and the study selection procedure are shown in Figure 1. A total of 4170 articles were identified in the original searches. After examining title and abstract, 90 were selected for a full-text review. After this phase, 31 papers were excluded: 2 because they did not include a proper measure of subjective aging, 3 due to not including an objective measure of cognition or cognitive decline, 16 since there was not enough information about measures employed and results obtained and the authors did not provide it after requesting, 6 because the publication was either a master’s thesis, PhD thesis, poster or conference abstract and the published article was already included in this review, 2 since they did not report the statistical results for the link between subjective aging and cognition and 3 because the full text was not available in English or Spanish. A total of 59 publications were included after the final review round. The complete characteristics of the 59 publications included are shown in Table 1.

### 3.2. Participant Characteristics and Sampling

Regarding publication type, forty-nine of the studies included were articles, three were conference abstracts reported in scientific journals, three were master theses, three were PhD theses, and one was a poster. In terms of design, 35 publications were longitudinal, and 25 were cross-sectional. One of the publications followed a micro-longitudinal design with a duration of nine days and was considered cross-sectional in this review. Only seven studies reported specific analyses regarding differences between age subsamples. Quality assessment was made for all publications included, available in Appendix A. Fifty-three studies were rated as high quality and six as moderate quality. There were no publications rated as low quality.

The majority of studies (44) took data from representative databases for the elderly population, such as the Health and Retirement study (15), the Midlife in the United States study (six), the Interdisciplinary Longitudinal Study of Adult Development (five), the German Aging Survey (four), the Irish Longitudinal Study on Ageing (three), the English Longitudinal Study of Ageing (two), the PROTECT study (two), the Lothian Birth Cohort (one), the Activity and Function in the Elderly in Ulm (one), the Baltimore Longitudinal Study on Aging (one), the China Longitudinal Aging Social Survey (one), the Ageing in Spain Longitudinal Study (one), the National Health and Aging Trends Study (one), the Norwegian Survey of Health and Ageing (one), the Study on Global AGEing and Adult Health (one) and their own random and representative sample (one). Others employed data sources with a somewhat lesser degree of representation (one), such as the Dementia Literacy Survey (one), the Douglas Hospital Longitudinal Study of Normal and Pathological Aging (one), the Mindfulness and Anticipatory Coping Everyday study (one), the Subjective Cognitive Decline study (one) and the Population-based and Inspiring Potential Activity for Old-old Inhabitants study (one). Moreover, there was a proportion of publications that employed convenience samples (nine). One study combined a convenience sample with data from the Midlife in the United States study.

### 3.3. Effect of Subjective Aging on Cognition

The most frequent combination was the analysis of the link between subjective age and memory (16), followed by self-perceptions of aging and memory (9), subjective age and global cognition (6) and subjective age and executive functions (6).

From the 59 studies included in this review, 48 reported results for the relationship between subjective aging and objective cognition, 9 reported results for the link between subjective aging and cognitive decline and 2 reported analyses regarding subjective age and both cognitive function and decline.

A total of 42 studies that examined subjective aging and cognition reported at least one significant result supporting the hypothesized relationship, whereas 8 did not report any analysis that yielded a significant link. From these 43 studies, 25 reported significant results in all their main analyses, and 18 studies showed mixed results. Three studies reported differences regarding the subjective aging constructs and/or subdimensions employed, two in relation to cognitive domain differences and four due to the combination of both. Three studies reported mixed results after including covariables and mediators in their analysis, and one because of the combination of subjective aging dimensions and the inclusion of a mediator variable. There was also variability due to subsample analysis; one study reported it combined with covariate effects, one combined with subjective aging constructs and dimensions and two combined with cognitive domains. Additionally, one study reported a relationship in cross-sectional analysis that did not replicate longitudinally.

Analyzing the results obtained for specific combinations of subjective aging constructs and cognitive domains, the most replicated link found was between subjective aging and memory, with 15 out of 16 studies reporting a significant effect. A complete description of combinations of subjective aging and cognitive constructs and the results obtained are shown in Table 2.

### 3.4. Cognitive Domains and Measures

Most studies examined one (39) or two (13) cognitive constructs; it was rare to find reports that explored three or more cognitive variables at the same time (eight). Memory was the most common cognitive domain studied, being included in 29 reports, followed by global cognition (17) and cognitive decline (11). Word list recall (24) was the most frequent memory measure, the MMSE (8) for global cognition, and the TICS (6) for cognitive decline. Seven studies examined speed of processing, seven executive functions, six reasoning, four verbal fluency, four attention, three language, three fluid intelligence, two crystallized intelligence, two visuospatial abilities, two working memory and one a composite of processing speed, attention and executive functions. A complete description of cognitive domains can be found in Table 3, which accounts for considerable variability in the measures employed.

### 3.5. Subjective Aging Constructs and Measures

There was a high variability in subjective aging constructs and measures employed. Overall, 49 studies examined only one subjective aging construct, whereas 8 explored two constructs, 1 studied three variables, and 1 studied four variables. The most frequent construct was subjective age (26), followed by self-perceptions of aging (15), attitudes towards own aging (6), attitudes to aging (5) and awareness of age-related change (5). Aging satisfaction and age stereotypes were included in two studies, and aging expectations, age identification and look age were each examined in only one study.

Regarding the measures employed, all the studies that included subjective age (26) employed a composite score based on the age felt and the participant’s age, although this measure was also employed in one study to measure views of aging. The second-most used measure was the unidimensional PGCMS (19), which was employed to evaluate attitudes towards own aging (seven), self-perceptions of aging (seven), age beliefs (three), satisfaction with aging (one), age stereotypes (one) and views on aging (one). Concerning multidimensional measures, the studies (six) that employed a variant of the AARC questionnaire studied awareness of age-related except for one, the studies (four) that employed any APQ variant measured self-perceptions of aging, and the studies (three) that used the AAQ explored attitudes to aging. One study studied self-perceptions of aging by using PEAS, and another by employing Age-Cog. One study used an ad hoc measure for age identification, and one used an ad hoc measure for look age.

Ten out of eleven studies that explored the link between subjective aging and cognitive decline reported at least a positive result in the expected direction. Seven showed positive results in all their main analyses, whereas three of them obtained mixed results because of covariates and mediators (one), differential effects between subjective aging constructs (one) and subsample variability (one).

### 3.6. Moderator Variables

Only nine studies included a moderation or mediation analysis. Seven studies that explored objective cognition included it, with a high variability in the constructs selected. Two studies included physical activity and depression. Activities of daily living, self-rated health, loneliness, smoking, alcohol consumption, social network, ageism, body mass index, learning self-efficacy, education, biomarkers, and social comparisons were each analyzed in one study. The only study regarding cognitive decline included leisure activity and control beliefs as moderators.

## 4. Discussion

Maintaining healthy cognition and reducing the incidence of cognitive impairment have become increasingly important due to population aging [110]. This systematic review offers a synthesized view of the role of subjective aging on cognition and cognitive decline. There is a growing body of studies in this field, with 59 studies identified that were mostly published in the last ten years. The majority of results indicate a significant relationship between these constructs. More-positive self-perceptions of aging, younger subjective age and positive attitudes towards own aging are related to better cognition and reduced risk of developing cognitive decline or dementia. Moreover, there was consistency among reports, such as the link between memory and subjective age and self-perceptions of aging on global cognition. However, there were major sources of variation that need to be considered.

The first specific aim was to examine possible differences between subjective aging constructs. In this regard, most reports examined subjective age, a construct that reflects a discrepancy between felt age and chronological age. Even as this measure has been employed systematically in this field of study and is a significant predictor of cognition [20] and health and longevity [31], its unidimensional nature might reflect an oversimplified picture of subjective aging that does not account for variation regarding representation content [36]. In this sense, several studies were identified that employed a multidimensional construct, such as attitudes towards own aging and self-perception of aging, which might be best suited to accounting for the complexity of this link. Additionally, some of these studies reported mixed results due to differential predictions and relationships, depending on the subscale employed. For example, in the case of awareness of age-related changes, Voelkner & Caskie [55] reported a link between cognitive losses and total losses but not of cognitive gains or total gains on memory, Zhu & Neupert [68] showed that total losses but not total gains are related to reasoning and Sabatini et al. [36] reported that cognitive losses and total losses but not gains were related to working memory. In the case of self-perceptions of aging, Robertson et al. [93] indicated that positive control and negative control are related to verbal fluency, whereas the timeline subscale was associated with prospective memory. This effect may also interact with chronological age, as Jung [94] pointed out that the social loss dimension of self-perceptions of aging predicts cognition only in a subsample of elderly adults, and Sabatini et al. [36] indicated that the cognitive gains dimension of awareness of age-related change predicted working memory and reasoning only in middle-aged and young old participants. All this evidence leads to the notion that some dimensions are best suited to accounting for certain effects on cognition.

Another objective was to identify the measures employed to address multidimensional constructs of subjective aging. Whereas some studies employed complex questionnaire measures, there was a high number of reports that employed the PGCMS for this task. It has been pointed out that this scale might not be best suited because of its unidimensional nature and unspecific item content [37]. Moreover, there was a significant variability regarding the construct measured, having been employed to explore attitudes towards own aging, self-perceptions for aging, age beliefs, satisfaction with aging, age stereotypes and views of aging. In order to increase the clarity and specificity of the umbrella concepts under the term of subjective aging and to better operationalize the measures employed, it is relevant to reduce this variability and commence using validated scales that respect the dimensionality and content of the constructs. Therefore, we recommend the use of the PGCMS only when assessing morale from a unidimensional point of view and the use of well-established questionnaires such as the APQ and the AARC for subjective aging.

The third aim was to examine variation regarding the cognitive constructs evaluated. Some of them seem to not replicate to the full extent along the reports, such as subjective age on global cognition and self-perceptions of aging on memory. Examination of studies that included more than one cognitive domain also revealed mixed outcomes; self-perception of aging seemed to predict executive function and memory, but not attention [92], and subjective age predicted memory but not executive function [82]. There can be several reasons for this variation. First, there is substantial variability in the neuropsychological measures used to assess cognition. For instance, executive functions were measured by the single use or combination of nine tasks that might reflect very distinct subcomponents, from category fluency to working memory and reasoning. This reasoning might also apply to language since it was operationalized with vocabulary, naming and fluency tasks that differ in the processes involved. A more salient case is global cognition, which included a variety of 10 tasks, the most common of which was the MMSE. As it has been pointed out, this measure might not be best suited to addressing the variability of normal cognition since it only captures modest age-related changes when compared to other neuropsychological tasks [111]. Therefore, we suggest that a more process-based approach may be beneficial for this field since it permits accounting for specific subcomponents of cognition and uses, to a lesser extent, a general construct such as global cognition. A second source is the covariates controlled in statistical analysis. For example, Morris et al. [71] found that subjective age predicted memory, executive function, language and speed of processing but that the effect after including depression only remained significant for language and speed of processing. In this regard, it is especially important to include and statistically control potential confounding variables. Finally, there might be an interaction with age. Some of the reports found differences when analyzing age subsamples on global cognition [80,94], working memory [36], memory, reasoning [36,106] and speed of processing [106]. This suggests that subjective aging might affect specific cognitive domains more profusely depending on the sample age, especially if it matches the neurotypical cognitive trajectories.

This review has several limitations. The inclusion of both cross-sectional and longitudinal designs encompasses a great amount of available evidence, but the latter are more appropriate for drawing conclusions about cause and directionality. Even if the inclusion of high-quality cross-sectional designs is valuable for examining the relationship between subjective aging and cognition, their results are to be carefully interpreted. Moreover, it does not offer a meta-analytic approach to better understand the effect sizes.

We suggest that future studies focus on longitudinal designs that include measures with strong psychometrical properties to measure subjective aging. In this regard, efforts could be directed to extending the evidence about questionnaires that have shown promise, such as the AARC and the APQ, and to avoiding using others that cast doubts, such as the PGMCS. Additionally, it might be of special interest to examine the differential effects of such constructs on cognition and whether some of them show stronger associations. Regarding cognition, we suggest a more detailed and process-based neuropsychological approach that employs sound tasks for specific cognitive domains or systems. Therefore, employing ad hoc measures or screening tools of general cognition is to be avoided. This process-based approach also needs to account for the specific domain that is more salient in each neuropsychological task. For instance, the picture completion task might be better suited to assessing selective attention rather than reasoning. Additionally, it is of special interest to increase the available evidence for some domains that are key for adult cognition and are underrepresented in the studies included, such as speed of processing, visuospatial abilities and working memory. This proposal also applies to cognitive decline and dementia, being relevant to ensuring that the selection of participants of this population or the splitting of a sample between people with and without cognitive decline follows the international criteria. Finally, we recommend including potential moderators for this relationship since only nine studies directly addressed it, and there was high variability in the moderator variable included. It is relevant to replicate the available evidence of studied moderators such as ageism [55], depression [71], loneliness [62], leisure activities [87] and activities of daily living and social network [78], and also to include other relevant variables such as optimism [17].

## 5. Conclusions

Subjective age is a relevant variable when predicting and explaining cognition and cognitive decline in old age. More-positive self-perceptions and attitudes towards own aging and a younger subjective age are related to better cognition and lower risk of cognitive decline. However, several variation issues need to be addressed in future research. It is important to create evidence for each of the umbrella terms included under subjective aging since they might affect cognition differentially. Moreover, these constructs need to be measured with psychometric questionnaires with solid evidence about their content and structure validity, and the same scale should not be used for different constructs. A more precise and process-oriented neuropsychological approach may be of great benefit since the measure selection and the sample age are potential sources of variation. Additionally, it is important to increase the evidence regarding key cognitive domains for the elderly population, such as speed of processing and working memory. Finally, it is relevant to identify and test potential moderators in this relationship to better understand the pathways implicated.

## Figures and Tables

**Figure 1 healthcare-11-03115-f001:**
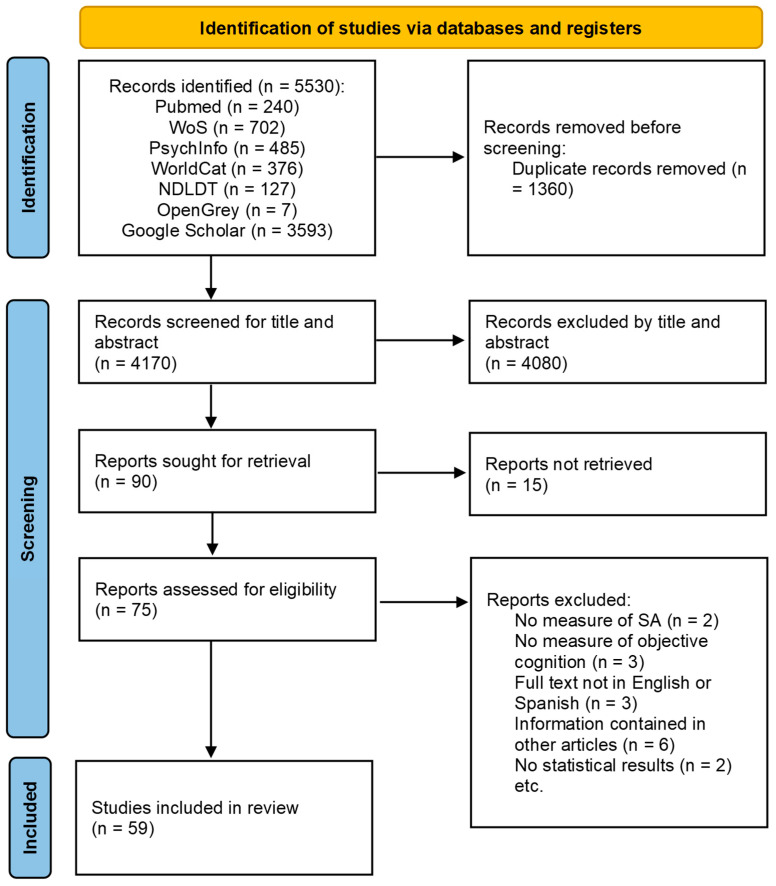
Flow diagram.

**Table 1 healthcare-11-03115-t001:** Full description of studies included.

Citation	Design	Sampling	Sample	SA	Cognition	Moderator	Covariate	Results
Voelkner & Caskie, 2023 [55]	Cross-sectional	Non-random	· 215 participants. · 66 years or elderly. · No dementia or cognitive impairment. · Living in the US. · Mean age of 69.06 (sd = 3.49) · 66% female	Awareness of age-related change (AARC-50)	Memory (word recall) Reasoning (word series, number series and letter series)	Ageism	Yes	There was a direct effect of cognitive losses (b = −0.17, *p* < 0.001) and total losses (b = −0.69, *p* < 0.001) on memory, as well as of cognitive gains (b = −0.12, *p* < 0.001), cognitive losses (b = −0.09, *p* < 0.001), total gains (b = −0.56, *p* < 0.001) and total losses (b = −0.39, *p* < 0.001) on reasoning. Ageism did not mediate this relationship.
Stephan et al., 2023 [56]	Longitudinal	HRS	· 2423 participants. · 50 years or elderly at baseline. · 60% female. · Mean age of 66.89 (sd = 9.22)	Subjective age (composite score of age discrepancy) Self-perceptions of aging (PGCMS)	Memory (word recall)	Biomarkers	Yes	There was a direct effect of subjective age (b = −1.13, *p* < 0.001) and self-perceptions of aging (b = −0.27, *p* < 0.001), and biomarkers mediated it.
Langballe et al., 2023 [57]	Cross-sectional	NORSE	· 817 participants. · 60 years or elderly. · 48.87% female. · 353 participants 60–69 years, 329 70–79 years, and 135 80 or elderly.	Subjective age (composite score of age discrepancy)	Global cognition (MoCA)	No	Yes	There was no significant relationship between subjective age and cognitive capacity.
Levy & Slade, 2023 [58]	Longitudinal	HRS	· 1716 participants. · 65 years or elderly. · Without dementia at baseline measurement. · 55.5% female. · Mean age of 77.8 years (sd = 7.5).	Age beliefs (PGCMS)	Cognitive decline (TICS)	No	Yes	Participants with MCI had 30.2% greater likelihood of recovering normal cognitive status if they held positive age beliefs and recovered 2 years faster.
Chapman et al., 2022 [59]	Cross-sectional	Parent study on subjective cognitive decline	· 136 participants. · Normal cognition and psychological status · 67.6% females. · Mean age of 73.69 (sd = 6.79).	Age stereotypes (ad hoc and IAT) Subjective age (composite score of age discrepancy) Age identification (ad hoc)	Memory (word recall)	No	Yes.	Age stereotype and age identification did not correlate with objective cognition nor explained a significant amount of variance over cognition in the regression analysis.
Fernández-Jiménez et al., 2022 [60]	Cross-sectional	ELES_PS	· 1124 participants. · Community-dwelling. ·50 years or elderly. · Living in Spain. · 54.4% female. ·Mean age of 64.84 (sd = 10.12).	Attitudes to aging (ad hoc) Self-perceptions of aging (ad hoc)	Global cognition (MMSE)	No	Yes	Self-perceptions of aging correlated with cognitive functioning (r = 0.173, *p* < 0.01), and its path was significant (Bca = 0.848). Cognitive functioning mediates the relationship between self-perceptions of aging and perceived health (B = 0.0032, SE = 0.002, *p* < 0.01).
Sabatini et al., 2022 [61]	Cross-sectional	PROTECT; 2019 sample.	· 6192 participants. · 50 years or elderly. · No dementia point. · 76% female. · Mean age of 66.1 years (sd = 7).	Awareness of age-related change (AARC-10)	Global cognition (digit span, self-ordered search, paired learning, and grammatical reasoning)	No	Yes	Class 1 (many gains, few losses) showed better cognition compared with class 2 (moderate gains, few losses), 3 (many gains, moderate losses) and 4 (many gains, many losses) in all four cognitive measures.
McGarrigle et al., 2022 [62]	Longitudinal	TILDA	· 4031 participants. · 50 years or elderly. · Subset analysis of elderly adults (65 years or elderly; n = 2359)	Self-perceptions of aging (APQ)	Global cognition (MMSE)	Depression, self-rated health, loneliness, smoking, physical function, alcohol consumption.	Yes	Participants within the highest tertile of negative APQ had a higher probability of being classified in the cognitive decline trajectory (RRR = 1.82 after controlling for covariates. This link was partially mediated (40%), notably by loneliness (22%) and poor self-rated health (7%). The result was similar in the 65 years or elderly subsample if loneliness became the strongest mediator.
Sabatini et al., 2022 [63]	Longitudinal	ILSE	· 103 participants. · Without dementia. · 50.5% females. · Mean age at 20-year follow-up of 82.5 years (sd = 1).	Self-perceptions of aging (AARC-50) Attitudes towards own aging (PGCMS)	Fluid intelligence (digit span, SDMT and block design) Crystallized intelligence (information, similarities, and picture completion)	No	Yes	A decline in digit symbol test score predicted fewer AARC gains (small effect size, R2 = 0.07) and higher AARC losses (small effect size, R2 = 0.03). However, change in cognition did not predict attitudes towards own aging.
Aftab et al., 2022 [64]	Cross-sectional	SAGE	· 1004 participants. · Community-dwelling · No dementia or terminal illness Age group 1 (21–39 years; n = 161), group 2 (40–59 years; n = 224), group 3 (60–79 years; n = 314) and group 4 (80+ years; n = 305).	Subjective age (composite score of age discrepancy)	Global cognition (TICS)	No	Yes	There was no correlation between age discrepancy and cognition in any age group.
Kaspar et al., 2022 [65]	Longitudinal	Non-random	· 912 participants. · With follow-up. · 80 years or elderly. · 50.24% female. · Mean age at baseline of 87 years (sd = 4.5)	Awareness of age-related change (AARC-10)	Global cognition (DemTect)	No	Yes	Intra-individual changes in cognition were not significantly associated with changes in AARC gains or losses.
Yuan et al., 2022 [66]	Longitudinal	Non-random	· 822 participants. · 65 years or elderly. · Cognitively healthy; 57.06% female. · Mean age of 70 (sd = 7).	Self-perceptions of aging (B-APQ)	Global cognition (MMSE)	No	Yes	There was a significant correlation between cognition and SPA (r = −0.175, *p* < 0.001).
Wahl et al., 2022 [67]	Longitudinal	BASE	BASE (1990/93 and 2017/18 cohort); 512 participants. Mean age of 77.	Views on aging (composite score of age discrepancy) Attitudes towards own aging (PGCMS).	Speed of processing (SDMT)	No	Yes	Positive attitudes towards aging correlated with speed of processing (r = 0.23, *p* < 0.05).
Zhu & Neupert, 2021 [68]	Cross-sectional	Mindfulness and Anticipatory Coping Everyday (MACE) study	· 112 participants. · 60–90 years old. · Living in the US. · Without cognitive impairment. · Mean age of 64.65 (sd = 4.86). · 56.3% female.	Awareness of age-related change (AARC-20)	Reasoning (letter series and item-number comparison) Memory (word recall)	No	Yes	There was an association between daily AARC losses and letter series scores (concurrent B = −0.09, lagged B = −0.09), but not with AARC gains. AARC gains or losses were not related to either word recall or number comparison.
Sabatini et al., 2021 [36]	Cross-sectional	PROTECT	· 6056 participants. · Cognitively healthy. · 76.2% female. · Mean age of 66 years (sd = 7). · 3111 participants were middle-aged (51–65 years); 2473 were in early old age (66–75 years) and 472 were in advanced old age (≥76 years).	Awareness of age-related change (cognitive AARC-50 subscale and AARC-10) Self-perceptions of aging (PGCMS) Subjective age (composite score of age discrepancy)	· Working memory (self-ordered search and digit span) · Reasoning (grammatical reasoning) · Memory (paired learning)	No	Yes	AARC gains and losses in cognition and AARC total losses predicted working memory and reasoning. AARC total gains predicted working memory and reasoning but not memory. Subjective age predicted working memory. AARC gains in cognition were predictors of cognition in the middle-aged and early old subsamples but not in the advanced old subsample. AARC losses in cognition predicted cognition in all subsamples.
Skoblow, 2021 [69]	Longitudinal	HRS	· 933 dyads. · 50 years or elderly. · 50% female. · Mean age for women of 63.91 (sd = 7.64). Mean age for men of 66.77 (sd = 7.79)	Self-perceptions of aging (PGCMS)	Memory (word recall)	No	Yes	There was no significant association between own or partner’s SPA and memory change at baseline or follow-up.
Stephan, et al., 2021 [70]	Longitudinal	HRS and MIDUS	· 2549 participants from HRS. No MCI at baseline; 60% female. Mean age of 69.66 (sd = 7.36). · 2499 participants from MIDUS. No MCI at baseline; 54% female. Mean age of 46.24 (sd = 11.25).	Subjective age (composite score of age discrepancy)	Memory (word recall, logic memory and brave man history) Visuospatial (constructional praxis and MMSE) Verbal fluency (category fluency) Speed–attention–executive (letter cancellation, SDMT, TMT a and B, stop and go) Reasoning (number series)	No	Yes	Elderly subjective age related to lower scores in episodic memory (MIDUS d = 0.14; HRS d = 0.24) and speed–attention–executive (MIDUS d = 0.25; HRS d = 0.33) in both samples. Elderly SA related to lower fluency (d = 0.3) and visuospatial ability (d = 0.25) in HRS sample, and the relationship between subjective age and episodic memory was stronger for elderly participants (B = −0.04) and participants with lower depression symptoms (B = 0.04) in HRS sample. After excluding participants with MCI, the relationships remained significant.
Morris et al., 2021 [71]	Cross-sectional	HRS	· 993 participants. · 65 years or elderly. · Without dementia. · 58.81% female. · Mean age of 75.85 (sd = 7.49)	Subjective age (composite score of age discrepancy)	Memory (word recall, logic memory and brave man history) Executive function (number series, TMT-B and visual reasoning) Visuospatial (constructional praxis) Speed of processing (letter cancellation and backwards count) Language (category fluency and naming)	Depression and chronological age	Yes	There was a direct effect of SA on episodic memory (b = 0.072, *p* < 0.01), executive functioning (b = 0.062, *p* < 0.01), language (b = 0.07, *p* < 0.25) and processing speed (b = −0.85, *p* < 0.001) but not visuoconstruction (b = 0.45, *p* > 0.05). Depression mediated 26.39% of the association between SA and episodic memory, 32.26% for executive functioning, 21.42% for language, and 23.35% for processing speed. However, after accounting for depression and covariates, only the effects of subjective age on language (b = 0.015, *p* = 0.27) and speed of processing (b = −0.2, *p* = 0.005) remained.
Schönstein et al., 2021 [72]	Longitudinal	ActiFE Ulm	· 526 participants. · With follow-up. · Age between 65 and 90 years at baseline. · 57% male.	Views of aging (PGCMS) Subjective age (composite score of age discrepancy)	Global cognition (MMSE)	No	Yes	General cognition did not significantly improve the prediction model for attitudes towards own aging (b = −0.04, *p* > 0.05) nor for subjective age (b = −0.03, *p* > 0.05).
Mariano et al., 2021 [73]	Longitudinal	HRS and DEAS	· Study 1: 3404 participants; 50 years or elderly. · Study 2: 4871 participants; 40 years or elderly.	Self-perceptions of aging (PGCMS)	Memory (word recall) Speed of processing (SDMT)	No	Yes	Study 1: Self-perceptions of aging correlated with cognition at T1 (r = 0.15, *p* < 0.001), T2 (r = 0.18, *p* < 0.001), T3 (r = 0.18, *p* < 0.001) and in the cross-lagged model (b = 0.09, *p* < 0.001). Study 2: Self-perceptions of aging correlated with cognition at T1 (r = 0.18, *p* < 0.001) and T2 (r = 0.22, *p* < 0.001) and in the cross-lagged model (Δχ2 (1) = 25.52, *p* < 0.001).
Qiao et al., 2021 [74]	Longitudinal	ELSA	· 6475 participants. · Community-dwelling. · 50 years or elderly. · Living in UK. · Without dementia at baseline. · Not outliers in subjective age (≥3 sd).	Subjective age (composite score of age discrepancy)	Memory (word recall) Executive function (category fluency) Cognitive decline (self-reported)	No	Yes	Elderly subjective age reported at baseline predicted poorer memory (β = −0.70, *p* = 0.02) and executive function (β = −1.56, *p* < 0.01) ten years later after controlling for covariates. Elderly subjective age was a risk factor for dementia (HR = 1.737) after controlling for covariates.
Stephan et al., 2021 [75]	Longitudinal	HRS	· 6341 participants. ·65 years or elderly. ·Without dementia. · Excluding outliers on gait speed and subjective age. · 57% female.	Subjective age (composite score of age discrepancy)	Cognitive decline (TICS)	No	Yes	Elderly subjective age participants were more likely to present MCR at baseline (OR = 4.44) and to develop MCR at follow-up (HR = 3.55).
Kisvetrová et al., 2021 [76]	Cross-sectional	Non-random	· PwD: 290 participants; 60 years or elderly, community-dwelling, diagnosed early-stage dementia. · PwoD: 209 participants; 60 years or elderly.	Attitudes to aging (AAQ)	Cognitive decline (MMSE)	No	Yes	PwD scored lower on all subscales of AAQ: Psychosocial loss (*p* = 0.001), physical change (*p* = 0.001) and psychological growth (*p* = 0.001).
Levy et al., 2020 [77]	Longitudinal	HRS	· 3895 participants. · 490 APOE carriers. · 60 years or elderly at baseline. · APOE ε2: APOE probability score >0.8. · 50.17% female. · Mean age of 71.07 (sd = 6.76).	Age beliefs (PGCMS)	Global cognition (TICS)	No	Yes	Positive age beliefs (F = 122.68, *p* < 0.001) and APOE ε2 (F = 7.87, *p* = 0.005) predicted better cognition. The interaction of positive age beliefs and APOE ε2 significantly predicted cognition (F = 7.74, *p* = 0.005).
Wang et al., 2020 [78]	Cross-sectional	China Longitudinal Aging Social Survey (CLASS)	· 8723 participants. · 60 years or elderly. · Completed MMSE and ATOA. · 46.3% female. · Mean age of 69.82 (sd = 7.54).	Attitudes towards own aging (PGCMS)	Global cognition (MMSE)	ADLs and social network (Lubben scale)	No	There was a partial mediation effect since cognition predicted attitudes towards own aging (b = 0.355) and social support (b = 0.15), and social support predicted attitudes towards own aging (b = −0.31).
Hajek, 2020 [79]	Longitudinal	DEAS	· 6348 participants. · Community-dwelling. · 50% female. · Mean age of 65 (sd = 10.6).	Aging satisfaction (PGCMS)	Global cognition (SDMT)	No	Yes	Decreases in cognitive functioning were associated with decreases in satisfaction with aging (β = 0.002) after controlling for all covariates.
Siebert et al., 2020 [80]	Longitudinal	ILSE	· Midlife age: 40 years or elderly at baseline. Mean age of 43.7 (sd = 0.92). · Old age: 60 years or elderly at baseline. Mean age of 62.5.	Attitudes towards own aging (PGCMS)	Global cognition (SDMT, digit span and block design)	No	Yes	Participants with positive attitudes towards own aging showed higher cognitive ability in both midlife (b = 0.25, *p* < 0.001) and old group (b = 0.41, *p* < 0.001). After controlling for covariates, this effect only remained in the midlife group.
Shao et al., 2020 [81]	Cross-sectional	Non-random	· 200 participants. · 60 years or elderly. · Complete measures. · No multivariate outliers. · Mean age of 65.42 (sd = 5.60). · 66.5% female.	Subjective age (composite score of age discrepancy)	Memory (word recall)	Learning self-efficacy and education (ad hoc)	Yes	Elderly subjective age was negatively associated with memory (r = −0.19). There was an indirect effect of subjective age on memory performance through learning self-efficacy (b = 0.04). There was an effect of elderly subjective age on learning self-efficacy (b = 0.18) and an interaction of subjective age and education on learning self-efficacy (b = 0.33).
Cerino et al., 2020 [82]	Cross-sectional	MIDUS	· 2621 participants. · 55.51%female. · Mean age of 64.06 (sd = 11.15).	Subjective age (composite score of age discrepancy)	Memory (word recall) Executive function (TICS)	No	Yes	Reporting a more youthful subjective age was associated with better episodic memory (est.−0.10, SE = 0.02, *p* < 0.001).
Choi et al., 2019 [23]	Cross-sectional	Dementia Literacy Survey	· 513 participants. · 60 years or elderly. · 55% female. · Mean age of 68.12 (sd = 5.65).	Subjective age (composite score of age discrepancy)	Global cognition (MMSE)	No	Yes	Subjective age correlated with cognitive functioning (r = 0.14, *p* = 0.01). In the full adjusted model, subjective age was a significant predictor of cognitive functioning (b = 0.08, *p* < 0.05).
Segel-Karpas & Palgi, 2022 [46]	Longitudinal	HRS	· 4624 participants. · 50 years or elderly at baseline. · Normal memory at baseline. · No dementia or stroke at follow-up.	Subjective age (composite score of age discrepancy)	Memory (word recall)	No	Yes	Subjective age correlated significantly with memory at t1 (r = −0.24, *p* < 0.001) and memory change (r = −0.15, *p* < 0.001).
Siebert et al., 2018 [83]	Longitudinal	ILSE	· T1: 1001 participants. Mean age of 62.5 (sd = 1). · T2 499 participants. Mean age of 66.6 (sd = 1.1). · T3 352 participants. Mean age of 74.3 (sd = 1.2).	Attitudes towards own aging (PGCMS)	Fluid intelligence (digit span, SDMT and block design) Crystallized intelligence (information, similarities, and picture completion)	No	Yes	After including covariables, more positive attitudes towards own aging baseline predicted less decline in fluid intelligence in men (βATOA = 0.71, *p* < 0.001) and explained 57% of the variance. However, it did not predict decline in women (βATOA = 0.7, *p* > 0.05). Positive attitudes did not predict less decline in crystallized intelligence.
Hughes & Lachman, 2017 [84]	Longitudinal	MIDUS	· 3427 participants. · Complete data in two waves. · Mean age of 55.92 (sd = 12.19).	Subjective age (composite score of age discrepancy)	Memory (word recall) Executive function (digit span backwards, category fluency, stop and go, and number series)	Social comparison	Yes	Direct effects showed that episodic memory was significantly related to cross-sectional subjective age (b = −0.42, *p* < 0.01), and indirect effects indicated that social comparisons mediated it (b = −0.09, κ2 = 0.01). There were no significant direct effects between longitudinal changes of subjective age and episodic memory or executive function.
Buggle, 2018 [85]	Longitudinal	ELSA	· Responded to both waves 4 and 7 of ELSA. · Wave 4: 11,050 participants; 64.15% female, mean age of 64.19 (sd = 8.57). · Wave 7: 9666 participants; 69.78% female, mean age of 69.66 (sd = 8.04).	Subjective age (composite score of age discrepancy)	Memory (word recall) Verbal fluency (category fluency and letter fluency)	No	Yes	Subjective age predicted immediate recall (β = −0.08; *p* < 0.001), delayed recall (β = −0.07; *p* < 0.001) and verbal fluency (β = −0.07; *p* < 0.001) at wave 4. Subjective age predicted immediate recall (β = −0.07; *p* < 0.001), delayed recall (β = −0.07; *p* < 0.001) and verbal fluency (β = −0.08; *p* < 0.001) at wave 7. Longitudinally, subjective age was a significant predictor of immediate recall (β = −0.03; *p* < 0.001), delayed recall (β = −0.03; *p* < 0.001) and verbal fluency (β = −0.01; *p*= 0.57).
Levy et al., 2018 [86]	Longitudinal	HRS	· 4756 participants. · 60 years or elderly at baseline. · No dementia at baseline. · APOE score >0.8. · Mean age of 72 years (sd = 7.19).	Age beliefs (PGCMS)	Cognitive decline (TICS)	No	Yes	Positive age beliefs were associated with lower risk of dementia in the total sample (RR = 0.81, 95% CI = 0.67, 0.97, *p* = 0.03) and in the APOE ε4 sample (RR = 0.69, 95% CI = 0.50, 0.94, *p* = 0.018).
Siebert et al., 2018 [87]	Longitudinal	ILSE	· 260 participants. · 60 years or elderly at baseline. · Cognitively healthy and without psychiatric disorders at baseline.	Attitudes towards own aging (PGCMS)	Cognitive decline (diagnosis or formal neuropsychological assessment)	Leisure activity and control beliefs	Yes	More negative ATOA at baseline was related to higher risk of MCI or AD at time 3 (b = 0.283, *p* < 0.05; OR 1.43). There was a significant path from ATOA to overall activity (b = 0.44, *p* < 0.001) but not from overall activity to future cognitive status. Adding cognitive leisure activity to the model weakened the direct effect of ATOA on cognitive status (b = 0.19, *p* = 0.09). There was a significant path from ATOA to external control beliefs (b = 0.17, *p* < 0.05) and external control beliefs to cognitive status at T3 (b = 0.15, *p* < 0.05) and a nonsignificant effect from ATOA to cognitive status (b = 0.18, *p* =0.08).
Stephan et al., 2018 [88]	Longitudinal	NHATS	· 4262 participants. · 65 years or elderly. · No dementia at baseline. · No subjective age outliers (≥3 sd). · Mean age of 76 (sd = 7.2).	Subjective age (composite score of age discrepancy)	Cognitive decline (diagnosis or formal neuropsychological assessment)	No	Yes	After controlling for demographics, depression and physical health, subjective age was not related to likelihood of developing dementia (HR = 1.11, *p* > 0.05). However, it was related in a subsample that excluded cases developed during the first year after the baseline measures (HR = 1.62, *p* < 0.05).
Tyrrell, 2017 [89]	Cross-sectional	HRS	· 361 participants. · 60 years or elderly. · 56% female. · Mean age of 73.67 (sd = 6.58).	Aging satisfaction (ad hoc) Aging expectations (ad hoc)	Memory (word recall) Language (vocabulary)	No	Yes	Aging satisfaction correlated with vocabulary (r = 0.12, *p* < 0.001) and memory (r = 0.15, *p* < 0.001), but it did not improve their predictive models (vocabulary: b = −0.004, *p* = 0.37; memory: b = 0.001, *p* = 0.73). Aging expectations correlated with vocabulary (r = 0.04, *p* < 0.001) and memory (r = 0.13, *p* < 0.001), but it did not improve their predictive models (vocabulary: b = −0.002, *p* = 0.11; memory: b = 0.000, *p* = 0.73).
Seidler & Wolff, 2017 [22]	Longitudinal	DEAS	· Baseline: 8198 adults. · Follow-up. · 49% female. · Mean age of 62.56 (sd = 11.93).	Self-perceptions of aging (Age-Cog)	Speed of processing (SDMT)	No	Yes	Physical loss (r = 0.24, *p* < 0.001) and personal growth (r = −0.17, *p* = 0.007) were correlated with speed of processing at baseline, but the cross-lagged paths of physical loss (Δχ^2^ = 0.12, *p* = 0.73; b = –0.03, *p* = 0.03) and personal growth (Δχ^2^ = 0.05, *p* = 0.83; b = 0.03, *p* = 0.01) on processing speed were equal without significant loss.
Stephan et al., 2017 [90]	Longitudinal	HRS	· 5772 participants. · 65 years or elderly at baseline. · No cognitive impairment at baseline. · No subjective age outliers (≥3 sd). · 59% female. · Mean age of 73.69 (sd = 6.24).	Subjective age (composite score of age discrepancy)	Cognitive decline (TICS)	No	Yes	An elderly subjective age at baseline was related to an increased likelihood of dementia (OR = 1.27, *p* < 0.001) and cognitive impairment (TICS-m <12: OR = 1.16, *p* < 0.001; TICS 7–11: OR 1.15, *p* < 0.001) after controlling for covariates, time interval, and baseline cognition.
Jaconelli et al., 2017 [91]	Cross-sectional	Non-random	· Dementia group: · France: 49 participants diagnosed with mild-to-moderate dementia of Alzheimer’s type aged 73–93 years old (MoCA: mean = 15.96, SD = 3.60). · US: 30 participants with dementia aged 77–82 years (MoCA: mean = 10.70, SD = 5.11). · Control group: 31 participants. No dementia; 60 years or elderly.	Subjective age (composite score of age discrepancy)	Cognitive decline (MoCA)	No	Yes	France: No significant difference in subjective age between the dementia and control groups after controlling for covariates (F = 0.06, *p* = 0.80). US: No significant difference in subjective age between the dementia and control groups after controlling for covariates (F(1, 54) = 0.56, *p* = 0.46). Both subsamples: No significant difference in subjective age between the dementia and control groups after controlling for covariates (d = 0.03; *p* > 0.05).
Robertson & Kenny, 2016 [92]	Cross-sectional	TILDA	· 4135 participants. · 50 years or elderly. · Community-dwelling. · No dementia, antidepressants, dementia medication or stroke history. · 53.4% female. · Mean age of 62 (sd = 8.7).	Self-perceptions of aging (B-APQ)	Global cognition (MMSE and MoCa) Executive function (visual reasoning, TMT-B and category fluency) Memory (picture recall) Attention (TMT-A)	No	Yes	Negative perceptions were a significant predictor of global cognition (B = −0.09, *p* < 0.05), executive function (B = −0.12, *p* < 0.001) and memory (B = −0.11, *p* < 0.01) but not attention (B = 0.03, *p* > 0.05).
Robertson et al., 2016 [93]	Longitudinal	TILDA	· 5896 participants. · 50 years or elderly at baseline. · Community-dwelling. · No stroke, Parkinson’s disease, MMSE < 18 or suspected dementia at baseline or in the intervening 2 years between waves. · 54.65% female. · Mean age of 63.17 (sd = 9.36).	Self-perceptions of aging (B-APQ)	Verbal fluency (category fluency) Memory (word recall and prospective memory)	No	Yes	Cross-sectional: Verbal fluency was associated with positive control (IRR = 0.38, *p* < 0.01) and negative control and consequences (IRR = −0.33, *p* < 0.05). Delayed memory was associated with positive control (IRR = 0.12, *p* < 0.01) and negative control and consequences (IRR = −0.21, *p* < 0.01). Immediate memory was associated with timeline (IRR = −0.13, *p* < 0.05), positive consequences (IRR = 0.12, *p* < 0.05), positive control (IRR = 0.15, *p* < 0.05) and negative consequences and control (IRR = −0.22, *p* < 0.01). · Longitudinal: Positive control was associated with animal naming in wave 2 (B = 0.43, *p* < 0.001). Negative control and consequences were associated with animal naming in wave 2 (B = −0.51, *p* < 0.001). Timeline was associated with the first prospective memory task (IRR = 0.98, *p* < 0.05).
Jung, 2016 [94]	Longitudinal	DEAS	· 2545 middle-aged participants (age between 40 and 64 years). ·1489 old participants (65 years or elderly).	Self-perceptions of aging (PEAS)	Global cognition (SDMT)	No	Yes	· Physical loss predicted changes in cognitive performance in the middle-aged group (T1→T2: –0.12, *p* < 0.01 and T3→T4: –0.12, *p* < 0.01) and the old group (T1→T2: –0.17, *p* < 0.01; T3→T4: –.26, *p* < 0.05). Cognitive performance only predicted changes in physical loss in the old group (T2→T3: –0.22, *p* < 0.05). · Social loss did not predict changes in cognitive performance in the middle-aged group, but it did in the old group (T1→T2: –0.30, *p* < 0.01; T3→T4: –0.40, *p* < 0.01). Moreover, cognitive performance predicted changes in social loss in the old group (T3 to T4: –0.51, *p* < 0.01). · Continuous growth predicted changes in cognitive performance in the middle-aged group (T1→T2:.13, *p* < 0.01) and the old group (T1→T2: 0.23, *p* < 0.01). Cognitive performance only predicted changes in continuous growth in the middle-aged group (T2→T3: 0.10, *p* < 0.05).
Stephan et al., 2016 [95]	Longitudinal	HRS	· 5809 participants at baseline; 3631 participants with complete data. · 50 years or elderly at baseline. · No subjective age outliers (= + 3 sd).	Subjective age (composite score of age discrepancy)	Memory (word recall)	No	Yes	A younger subjective age at baseline was related to better immediate recall (β = 0.05, *p* < 0.001), delayed recall (β = 0.05, *p* < 0.001) and total memory (β = 0.05, *p* < 0.001) at baseline, and with better immediate recall (β = 0.04, *p* < 0.001), delayed recall (β = 0.03, *p* < 0.01) and total memory (β = 0.04, *p* < 0.001) longitudinally. Depressive symptoms mediated the association between subjective age and changes in immediate recall (β = 0.04, *p* < 0.001), delayed recall (β = 0.05, *p* < 0.001) and total memory (β = 0.08, *p* < 0.001).
Hagood & Gruenewald, 2015 [96]	Longitudinal	HRS	·1518 participants · Age between 65 and 99.	Self-perceptions of aging (PGCMS)	Memory (word recall)	No	No	Negative self-perceptions of aging are related to reductions of memory over time (β = −0.26; *p* < 0.001) after controlling for covariables.
Hülür et al., 2015 [97]	Longitudinal	HRS	· 5824 participants at T1. · 50 years or elderly. · 58% female. · Mean age of 64.27 (sd = 9.9).	Subjective age (composite score of age discrepancy)	Memory (word recall)	No	No	Subjective age predicted correlated with memory at baseline (r = −0.31; *p* < 0.01) but did not improve the predictive model (b = −0.01, *p* > 0.01).
Ihira et al., 2015 [98]	Cross-sectional	Population-based and Inspiring Potential Activity for Old-old Inhabitants (PIPAOI) study	· 275 participants. · 75 years or elderly. · No recent hospitalization, stroke, cardiovascular disease, diabetes, osteoporosis, dementia, depression, or schizophrenia. · 59.1% female. · Mean age of 80 (sd = 4.1).	Subjective age (composite score of age discrepancy)	Attention (TMT-A) Executive function (TMT-B) Speed of processing (SDMT) Memory (word recall)	No	Yes	Subjective cognitive age correlated with attention (r = 0.12, *p* < 0.01), executive function (r = 0.18, *p* < 0.01), speed of processing (r = −0.29, *p* < 0.01) and memory (r = −0.25, *p* < 0.01). Subjective physical age only correlated with speed of processing (r = −0.25, *p* < 0.01). Word list score was a significant predictor of subjective cognitive age (OR = 1.26, *p* = 0.03).
Chasteen et al., 2015 [99]	Cross-sectional	Non-random	· 301 elderly adults. · 63.78% female. · Mean age of 71.13 (sd = 7.4).	Views of aging (ARS)	Memory (word recall)	No	No	There was no correlation between views of aging and memory (r = −0.07, *p* > 0.05).
Shenkin et al., 2014 [100]	Longitudinal	Lothian Birth Cohort (1936)	· 1091 participants. · Community-dwelling.	Attitudes to aging (AAQ)	Fluid intelligence (letter number, digit span backwards, matrix reasoning, block design, SDMT and symbol search) Global cognition (MMSE)	No	Yes	Fluid intelligence correlated with the psychosocial loss (r = −0.13, *p* < 0.001) and physical change (r = 0.127, *p* < 0.001) subscales of the AAQ. Fluid intelligence did not predict scores of psychosocial loss (b = −0.014, *p* > 0.05), physical change (b = 0.068, *p* > 0.05) or psychological growth (b = 0.009, *p* > 0.05) subscales of the AAQ.
Stephan et al., 2014 [101]	Longitudinal	MIDUS	· 1368 participants. · 50 years or elderly at baseline. · Without neurological disorders. · 76% female. · Mean age of 59.95 (sd = 6.73).	Subjective age (composite score of age discrepancy)	Memory (word recall) Executive function (digit span backwards, category fluency, stop and go, and number series)	Body mass index and physical function	Yes	· Subjective age correlated significantly with episodic memory (r = 0.06, *p* < 0.05) and executive function (r = 0.06, *p* < 0.05). Subjective age was a significant predictor of episodic memory (b = 0.05, *p* < 0.05) and executive function (b = 0.05, *p* < 0.05). BMI partially mediated the relationship between subjective age and episodic memory (indirect effect b = 0.07). Physical activity partially mediated the relationship between subjective age and executive function (indirect effect b = 0.02).
Hughes, 2014 [102]	Cross-sectional	1A and 1B: Own sample, non-randomized. 1C: MIDUS	· 1A: 47 participants. Scores equal to or higher than 26 at MMSE; 65 years or elderly; 53.17% female. Mean age of 71.4 (sd = 7.1). · 1B: 78 participants. Scores equal to or higher than 26 at MMSE; 55 years or elderly. Mean age of 59.87 (sd = 4.21). · 1C: 3228 participants; 54.3% female. Mean age of 55.92 (sd = 12.16).	Subjective age (composite score of age discrepancy)	Memory (word recall) Language (vocabulary) Speed of processing (lexical decision and backwards count) Working memory (digit span and digit span backwards) Verbal fluency (category exemplar and category fluency) Reasoning (number series) Attention (backwards count)	No	Yes	1A: No correlations between subjective age and cognitive indicators after controlling for chronological age. Applying bootstrap, it was correlated with recall (r = 0.11, *p* = 0.00), vocabulary (r = 0.07, *p* = 0.03), category fluency (r = −0.13, *p* = 0.00), F-A-S perseverative errors (r = −0.09, *p* = 0.01) and category fluency perseverative errors (r = −0.11, *p* = 0.00). 1B: Baseline subjective age was correlated with reasoning ability after controlling for chronological age (r = −0.26, *p* = 0.02). Applying bootstrap, subjective age correlated with processing speed (r = 0.07, *p* = 0.03), reasoning ability (r = −0.14, *p* = 0.00), category fluency (r = −0.10, *p* = 0.00), LDT prediction (r = −0.13, *p* = 0.00), reasoning ability prediction (r = 0.13, *p* = 0.00) and vocabulary prediction (r = −0.14, *p* = 0.00). 1C: Baseline subjective age correlated with immediate recall (r = −0.09, *p* = 0.00), delayed recall (r = −0.09, *p* = 0.00), the proportion of forgotten words between the two recall tests (r = 0.05, *p* = 0.03), processing speed (r = −0.06, *p* = 0.00), working memory (r = −0.06, *p* = 0.01), reasoning ability (r = −0.05, *p* = 0.03) and category fluency (r = −0.07, *p* = 0.00).
Levy et al., 2012 [103]	Longitudinal	BLSA	· 1st hypothesis: 395 participants. Community-dwelling. At least 22 years old at baseline; 28.6% female. Mean age of 45 years at baseline. · 2nd hypothesis: 87 participants. Measure of self-relevance; 40 years or elderly; 31.03% female. Mean age of 53 years at baseline.	Age stereotypes (PGCMS) Self-relevance (ad hoc)	Memory (picture recall)	No	Yes	After controlling for covariates, there was a link between age stereotypes and memory over time (b = −0.24, *p* = 0.04, d = 2) that increased as the participants aged. Moreover, there was a significant interaction between age stereotypes and self-relevance on memory (B = −31.10, *p* = 0.0002, d = 3.7).
Sindi, 2014 [104]	Cross-sectional	Douglas Hospital Longitudinal Study of Normal and Pathological Aging	· 40 participants. · With no history of head trauma, cerebral vascular accident, alcohol abuse, use of medications or use of anesthesia in the previous year. · Cognitively healthy. · 55% female. · Mean age of 71.25 (sd = 1.39).	Self-perceptions of aging (PGCMS)	Memory (paired learning)	No	Yes	Self-perceptions of aging did not predict total related (*p* = 0.718) nor total unrelated (*p* = 0.544) word pairs recall.
Trigg et al., 2012 [105]	Cross-sectional	Non-random	·PwD: 56 participants. Mild dementia. · PwoD: 84 participants; 60 years or elderly. Community-dwelling. Cognitively healthy.	Attitudes to aging (AAQ)	Cognitive decline (MoCA)	No	No	There were significant differences in the psychosocial loss subscale of the AAQ (t = 3.56, *p* < 0.01).
Paggi et al., 2011 [106]	Cross-sectional	ILSE	· Middle-aged: 501 participants. Mean age of 44.2. · Elderly: 499 participants. Mean age of 62.9.	Self-perceptions of aging (PGCMS)	Reasoning (block design, spatial capacity and picture completion) Speed of processing (D2, SDMT and connect the numbers) Memory (digit span, word recall and picture recall)	No	No	Aging self-perceptions were not related to speed of processing (Es = 0.02, *p* > 0.05), memory (Es = 0.02, *p* > 0.05) or reasoning (Es = 0.00, *p* > 0.05) in the middle-aged sample. Aging self-perceptions were related to memory (Es = 0.07, *p* < 0.05) and reasoning (Es = 0.09, *p* < 0.05) in the elderly adults sample, but processing speed did not mediate this relationship.
Lee & Hong, 2010 [107]	Cross-sectional	Non-random	· 1345 participants. · 60 years or elderly. · Without cognitive impairment. · 74.02% female. · Mean age of 75.8 (sd = 6.2).	Subjective age (composite score of age discrepancy)	Global cognition (MoCA)	No	Yes	There was a significant difference in the estimated marginal means of MMSE among 4 quartile groups after controlling for covariates (F = 13.122, *p* < 0.0001). Subjective age was associated with general cognition in the elderly after adjusting for covariates (b = 0.116, *p* < 0.0001).
Murphy, 2009 [108]	Longitudinal	MIDUS and BOLOS	· MIDUS: 4955 participants at follow-up; 53.34% female. Mean age of 55.45 (sd = 12.44). · BOLOS: 151 participants at follow-up; 38.41% female. Mean age of 46.57.	Subjective age (composite score of age discrepancy) Look age (composite score of age discrepancy)	Global cognition (BTACT)	No	Yes	The young feel age group obtained higher scores in cognition (F = 17.09, *p* < 0.001) when compared with the same and old feel age group.
Levy & Langer, 1994 [109]	Cross-sectional	Non-random	· 90 participants. · From the American deaf, American hearing and Mainland China. · Participants were 45 young adults (mean age = 22) and 45 elderly adults (mean age = 70).	Attitudes to aging (FAQ)	Memory (picture recall and paired learning)	No	Yes	There was a significant correlation between views of aging and memory in the old group (r = 0.49, *p* < 0.01). The direct path between views of aging and memory was significant (ES = −0.31, *p* < 0.001). The direct path between positive culture and memory was not significant (*p* > 0.05). The direct paths between age and memory and age and positive age views were not significant (*p* > 0.05).

**Table 2 healthcare-11-03115-t002:** Cognitive domains studied by subjective aging construct.

Cognitive Domain	Subjective Aging Construct											
	Subjective Age	SPA	ATOA	Attitudes to Aging	Awareness of Age-Related Change	Age Beliefs	Views of Aging	Aging Satisfaction	Age Stereotypes	Aging Expectations	Age Identification	Look Age
Memory	16 (15)	9 (6)		1 (1)	3 (2)		1 (0)	1 (0)	2 (1)	1 (0)	1 (0)	
Global cognition	6 (3)	6 (5)	2 (2)	3 (2)	3 (1)	1 (1)	1 (0)	1 (1)				1 (0)
Cognitive decline	4 (3)		1 (1)	2 (2)		2 (2)						
Speed of processing	3 (3)	3 (1)	1 (1)				1 (0)					
Executive functions	6 (4)	1 (1)										
Reasoning	3 (1)	2 (1)			3 (3)							
Attention	2 (1)	1 (0)										
Language	2 (1)							1 (0)		1 (0)		
Fluid intelligence		1 (1)	2 (1)	1 (1)								
Crystallized intelligence		1 (0)	2 (0)									
Visuospatial	2 (1)											
Working memory	2 (2)	1 (0)			1 (1)							
Verbal fluency	3 (3)	1 (1)										
Speed–attention–executive	1 (1)											
Total	50 (35)	26 (16)	8 (5)	7 (6)	10 (7)	3 (3)	3 (0)	3 (1)	2 (1)	2 (0)	1 (0)	1 (0)

Note: Each cell reports the number of studies that analyzed the combination of subjective aging and cognitive domains, and the number of positive results is in brackets. Subjective aging constructs and cognitive domains are reported as in the original studies. SPA: self-perceptions of aging, ATOA: attitudes towards own aging. For a comprehensive review of subjective aging constructs, see Diehl & Wahl (2010) [30].

**Table 3 healthcare-11-03115-t003:** Cognitive domains and measures.

Domain	N of Studies Included	Measure	N of Times Employed
Memory	29		
		Word recall	24
		Picture recall	4
		Brave Man	2
		Digit span	1
		Paired learning	4
		Prospective memory task	1
		Logical memory	1
Global cognition	17		
		MMSE	8
		SDMT	3
		TICS	2
		Digit span	2
		MoCA	2
		Self-ordered search	1
		Grammatical reasoning	1
		BTACT	1
		Block design	1
		DemTect	1
Cognitive decline	11		
		TICS	6
		MMSE	3
		Diagnosis or formal neuropsychological assessment	2
		MoCA	1
		Self-reported	1
Speed of processing	7		
		SDMT	6
		Backwards count	2
		TMT-A	1
		Letter cancellation	1
		D2	1
Executive function	7		
		Category fluency	4
		Number series	3
		TMT-B	3
		Number series	3
		Digit span (backwards)	2
		Stop and go	2
		TICS	1
		Visual reasoning	1
Reasoning	6		
		Number series	3
		Block design	1
		Spatial capacity	1
		Picture completion	1
		Grammatical reasoning	1
		Letter series	1
		Number comparison	1
Verbal fluency	4		
		Category fluency	4
		Letter fluency	1
Attention	3		
		TMT-A	2
		Backwards count	1
Language	3		
		Vocabulary	2
		Naming	1
		Category fluency	1
Fluid intelligence	3		
		SDMT	3
		Block design	3
		Digit span (backwards)	1
		Digit span	1
		Letter-number	1
		Visual reasoning	1
		Symbol search	1
Crystallized intelligence	2		
		Information	2
		Similarities	2
		Picture completion	2
Visuospatial	2		
		Constructional praxis	3
		Visual reasoning	1
Working memory	2		
		Digit span	2
		Self-ordered search	1
		Digit span (backwards)	1
Speed–attention–executive	1		
		Letter cancellation	1
		SDMT	1
		TMT-A	1
		TMT-B	1
		Stop and go	1

## Data Availability

The datasets generated during the current study are available from the corresponding author upon request.

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
