# Peer review of "The Relationship between Subjective Aging and Cognition in Elderly People: A Systematic Review"

_healthcare, 2023, doi:10.3390/healthcare11243115_

Round 1

Reviewer 1 Report

Comments and Suggestions for Authors

This review compiles a substantial amount of literature summarizing links between subjective ageing and cognition. I find the piece to be a useful summary with a broader scope than some existing reviews. I believe the article needs only minor corrections, mainly to increase the utility and citeability of the piece.

It is more aligned with literature to use the term ‘older adults’ as opposed to ‘Elderly’ etc. This should be changed throughout, especially in the title (e.g., search the word older in the reference list vs elder).

On line 67 the review claims to address dementia and MCI but these do not feature much in the discussion.

Could the abstract contain a theoretical point rather than methodological criticism as its take-home message. Perhaps a firm conclusion on the direction of the relationship between subjective ageing and cognition (which is currently softly worded in the middle of the abstract).

Table 2 could do with a final row of totals for each scale (e.g., ATOA would be 8 (5)) so one could compare the subjective ageing measures. Table 2 could do with a larger table note to explain some of the columns – from Age beliefs onwards, it is a bit unclear what is being measured.

I don’t think it is fair to highlight no links between crystallized intelligence and subjective ageing based on Table 2, with just 3 studies (line 244). This also features in the discussion around lines 343. Could more be done to reach this conclusion from the current data? If not, perhaps it should be removed.

I don’t necessarily expect a change to the manuscript for this, rather just an answer in your review response: Why does Table 3 not break down the measures into number that are linked to subjective ageing and number that are not linked to subjective ageing as is done in Table 2 with parenthesis? What does Table 3 add? It seems like very similar information to Table 2, could the tables be combined?

The utility of the paper could be increased, probably in the conclusion. A researcher reading this review might simply want to use the best measure of subjective ageing in a study of self-perceptions of ageing and cognition. It is argued on lines 341-342 that the APQ and AARC are promising. But this information is buried in the text and one would have to search for the acronyms to get to the citations.

The conclusion should also include a firmer message outlining which cognitive measures are more or less explored in the domain of subjective ageing and which need further investigation.

Comments on the Quality of English Language

Only minor issues. I had to read a few sentences twice towards the beginning of the introduction. There were some unwanted periods around citations.

Author Response

Dear reviewer:

I’d like to thank you for your time and effort reviewing this systematic review. I am pleased to inform you about the changes made in the manuscripts, that you will find highlighted in the updated version, as well as responding your comments:

  • The term “older” has been replaced for “elderly” along the document.
  • Several inclusions about the relationship between subjective aging and cognitive decline have been included, such as in the abstract (line 17) and discussion (lines 348 and 453).
  • A final row of totals has been included in table two. Regarding subjective aging constructs, there has been an addition in the note stating that they are the specific constructs reported by the original authors, as well as a suggested paper that specifically covers theoretical differences among them.
  • Following your suggestion, conclusions about crystallized intelligence are removed, since the sample of studies that addressed it is very reduced.
  • Regarding table 3, it is included with the intention of stating the high variability of cognitive measures that are employed to measure the same cognitive domain. Therefore, we consider that it adds some information about cognitive measurement variability and justifies the claim of employing a process-based approach. To better describe this idea, an additional brief explanation has been included at lines 439-442.
  • More clear conclusions have been included, as well as some prospective recommendations. The effect of subjective aging on cognition is now outlined in the discussion (lines 345-349) and conclusions (lines 450-453), as well as the necessity to explore cognitive domains that are underrepresented (lines 436-439 and lines 459-461).

Kind regards.

Reviewer 2 Report

Comments and Suggestions for Authors

Manuscript ID: healthcare-2727907

Manuscript title: The Relationship Between Subjective Aging and Cognition in Elderly People. A Systematic Review

I have reviewed your paper, which is an interesting study on The Relationship Between Subjective Aging and Cognition in Elderly People. A Systematic Review. Please consider the following comments regarding your review paper:

1-      Clarification of Terms:

Please provide a clearer definition and distinction of the various concepts encompassed under the term 'subjective aging'. Specifically, elaborate on how these concepts differentially impact cognitive functions.

2-      Choice and Validity of Measurement Scales:

Consider revising the selection of measurement scales, particularly the applicability of the PGCMS. Explore alternative scales that can more accurately measure the multifaceted aspects of subjective aging.

3-      Detailing the Neuropsychological Approach:

Elaborate on the specifics of the proposed process-oriented neuropsychological approach and its implications for cognitive functions. Provide detailed methodology and expected outcomes of this approach.

4-      Identification of Moderators:

Identify and analyze potential moderators that may influence the relationship between subjective aging and cognitive functions. Provide a detailed analysis of how these moderators could impact this relationship.

5-      Explanation of Sample Selection:

Provide a more comprehensive explanation of the criteria for sample selection and discuss how this selection may influence the study’s outcomes.

6-      Transparency in Statistical Analysis:

Enhance transparency and detail in the description of the statistical models and analytical methods used. This will strengthen the validity of your analysis and findings.

7-      Study Limitations and Future Research Directions:

Elaborate on the limitations of your study in greater detail and suggest specific directions for future research based on these limitations. This will provide valuable insights for subsequent studies in this field.

Comments on the Quality of English Language

Please provide a clearer definition and distinction of the various concepts encompassed under the term 'subjective aging'. Specifically, elaborate on how these concepts differentially impact cognitive functions.

Author Response

Dear reviewer:

I’d like to thank you for your time and effort reviewing this systematic review. I am pleased to inform you about the changes made in the manuscripts, that you will find highlighted in the updated version, as well as responding your comments:

  1. A clear description of potential differential effects of subjective aging construct is now included (lines 98-118).
  2. We thank you for the suggestion of considering the validity of the scales employed to measure subjective aging. In this regard, we believe that the PGCMS is not the tool best suited for this task, but it has been employed extensively in the field of subjective aging. Consequently, we include it in this review to account for all possible evidence. Nonetheless, a recommendation it is pointed out during the discussion to cease measuring self-perceptions of aging with this questionnaire.
  3. A more precise description of the neuropsychological approach proposed is included in the discussion (lines 430-442).
  4. A description of potential moderators has been included in the introduction, as well as the possible effect pathways (lines146-163).
  5. An extension of the sample selection criteria is included in lines 184-190.
  6. There is a brief addition to the methods employed to select the studies and data extraction (lines 195-196). Since there are no statistical procedures involved during this review, we hope that it accounts for your suggestion.
  7. An extensive limitations and future studies sections have been included in the discussion (lines 419 – 448).

Kind regards.

Round 2

Reviewer 2 Report

Comments and Suggestions for Authors

The authors have responded appropriately to peer review opinions. No additional comments are provided.